# A Minimalistic Prediction Model to Determine Energy Production and Costs of Offshore Wind Farms

**Jens Nørkær Sørensen** [1,*] and **Gunner Christian Larsen** [2]

1 DTU Wind Energy, Aero- and Fluid Dynamics, 2800 Lyngby, Denmark
2 DTU Wind Energy, Response, Aero-elasticity, Control and Hydrodynamics, 4000 Roskilde, Denmark; gula@dtu.dk
* Correspondence: jnso@dtu.dk; Tel.: +45-2091-1291

**Abstract:** A numerical framework for determining the available wind power and associated costs related to the development of large-scale offshore wind farms is presented. The idea is to develop a fast and robust minimal prediction model, which with a limited number of easy accessible input variables can determine the annual energy output and associated costs for a specified offshore wind farm. The utilized approach combines an energy production model for offshore-located wind farms with an associated cost model that only demands global input parameters, such as wind turbine rotor diameter, nameplate capacity, area of the wind farm, number of turbines, water depth, and mean wind speed Weibull parameters for the site. The cost model includes expressions for the most essential wind farm cost elements—such as costs of wind turbines, support structures, cables and electrical substations, as well as costs of operation and maintenance—as function of rotor size, interspatial distance between the wind turbines, and water depth. The numbers used in the cost model are based on previous but updatable experiences from offshore wind farms, and are therefore, in general, moderately conservative. The model is validated against data from existing wind farms, and shows generally a very good agreement with actual performance and cost results for a series of well-documented wind farms.

**Keywords:** offshore wind farms; resource assessment; cost modeling; levelized cost of energy

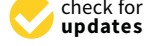

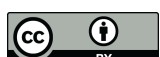

## 1. Introduction

Measured by wind resources as well as by the investments and efforts of the European wind energy industry to reduce the cost of offshore wind power, it is clear that offshore wind power will become a very important part of the future European wind power production. As an illustration of this, more than 100 offshore wind farms have been erected in Europe to date, contributing with an installed capacity of more than 18,000 MW, of which about one third have been established within the past two years [1]. An important question in relation to this is to what extent the oceans can be exploited with respect to a massive utilization of wind power and, associated with that, what are the economic aspects of doing this? To answer these questions it is required to determine the available wind resources as well as the associated costs of erecting and operating wind turbines in wind farms on the ocean. The installation of turbines in wind farms will, due to mutual wake effects, alter the local wind conditions. Hence, erecting wind turbines close to each other will reduce the internal wind farm wind speed and thus in turn the efficiency of the total power production. On the other hand, if the wind turbines are too far from each other, the full potential of the wind resources on the ocean will not be achieved, and the financial expenses related to the internal grid will increase significantly. An important parameter in this context is the average distance between the wind turbines, measured in rotor diameters, which is a reference length for wind farms. Today, in a typical wind farm, such as the Danish Rødsand or Horns Rev wind farms, the wind turbines are located about seven diameters from each other in order to diminish the wake effects. However, in other wind farms, e.g.,

the Lillgrund wind farm, this number may be substantially smaller. Another important parameter is the size of the wind turbines, measured in installed generator power and rotor diameter. While the size of wind turbines erected onshore has stabilized on a maximum of about 3.5 MW mainly due to logistic challenges, the size of offshore wind turbines is still increasing because of the influence of size on the reduction of cost of energy, which is much more pronounced offshore than onshore. Today, the biggest offshore wind turbines have a diameter of more than 200 m and an installed generator capacity of 8 MW. Yet an important parameter in a cost analysis of offshore wind turbines is the water depth, as the price of foundations and substructures heavily depends on the water depth. Therefore, a wind resource analysis requires it to be complemented with a bathymetric analysis to evaluate the economic potential of a given offshore site. Other important economic parameters are costs of installation as well as operation and maintenance (OM), both of which are substantially increased at offshore locations because of the harsh weather conditions appearing in the ocean.

A way to measure the economics of wind farms is to determine installation costs (capital expenditure or CAPEX), operation and maintenance costs (operational expenditure or OPEX), annual energy production (AEP), and levelized cost of energy (LCOE) for a range of different parameters, such as wind turbine size, average distance between the wind turbines, site wind climate, distance to the shore, and water depth. The LCOE, which is a commonly used measure to compare costs of different energy technologies, represents the cost over the lifetime of an investment compared to the expected energy production. Hence, it does not depend on the varying revenues or the production on a specific day or year.

Today, there exists a large amount of data related to the economics of wind farms, and there are already several models in use for cost analysis and for computing LCOE for offshore wind farms. In the past years, substantial experience of costs of wind farms has been gained based on published data of actual expenses and operational statistics. Some of this has been compiled and reported in technical reports and reviews (Ernst and Young [2], EWEA (European Wind Energy Association: Brussels, Belgium) "Wind Energy— The Facts" [3], Morthorst and Kitzing [4]). A recent review of Gonzalez-Rodriguez [5] gives a comprehensive overview of the most pertinent cost models, including historical data for the most important economic parameters, such as inflation rates and commodity prices. Some of the models aim at clarifying detailed costs of parts of wind turbines, such as Lundberg [6] and Castro-Santos [7], whereas others deal with simplifying global parametrizations to achieve parametrical studies on LCOE for understanding the key cost factors of wind farms (Ioannoua [8]). Aspects of floating wind turbines have also been modeled and analyzed, e.g., by Myhr et al. [9] and Bjergseter and Ågotnes [10], who compared the LCOE for bottom-fixed and floating wind farms based on a comprehensive cost analysis of a reference wind farm.

In recent years, various international collaborative projects have been conducted for analyzing different aspects and future trends of offshore wind power deployment. In the IEA (International Energy Agency: Paris, France) Wind Task 26 (Smart et al. [11] and Noonan [12]), a model based on a combination of bottom-up component modeling and higher-level industry data was employed to determine typical LCOE-values for offshore wind farms in different countries. Here the WAsP (Wind Atlas and Analysis and Application Program; www.wasp.dk) software of DTU Wind Energy at the Technical University of Denmark was used to determine the annual energy production, and the cost model used was based an internal spreadsheet delivered by one of the partners. In another study carried out under the auspices of the North Sea Wind Power Hub Consortium (Ruijgrok et al. [13]), a cost evaluation was carried out to assess the potential for the future wind energy exploitation of the North Sea. Here the annual energy production was determined using the "Farmflow" code from ECN (Energy Research Centre of The Netherlands: North Holland, The Netherlands), and the costs of a reference turbine were determined by calculating each component of the nacelle and blades using the research institution ECN

part of TNO's cost modeling tools and knowledge. Although these works have much valuable information for decision makers, the content of the integrated models used are not sufficiently transparent to be utilized by potential users outside the project consortia.

Besides a cost model, a fully integrated analysis also demands a wind resource model, which combines the prediction of site-dependent wind resources and the losses due to wake effects. For a thorough overview of the various wind farm flow models, we refer the reader to the review by Porté-Agel et al. [14]. Unfortunately, most of the methods described concern optimization issues and are either too computationally expensive or too complex to be employed as simple overall analysis models. One exception, however, is the fully developed wind farm array boundary layer model, which was originally developed by Templin [15] and Newman [16] to determine the impact of large-scale utilization of wind power on the available wind resources. The method was refined by Frandsen [17] and later validated against large eddy simulations by Calaf et al. [18], and extended to include atmospheric stability properties by Abkar and Porté-Agel [19]. The model has recently been coupled to simple cost models by Meyers and Meneveau [20] and Stevens et al. [21] in order to determine the optimal spacing of turbines in large wind farms on land.

The aim of the present work is to develop and validate a simple, robust, and reliable numerical framework that may aid developers and decision-makers in assessing the economic aspects of developing offshore wind power at a specific site. The idea is to have a tool to make initial cost analyses without the need for any detailed technical information regarding the utilized wind turbines or the actual topology of the wind farm. Hence, ideally, the model only requires information on turbine size (rotor diameter and nameplate capacity), size and intensity of wind farm (number of turbines per area unit, or, alternatively, average distance between the turbines), and site Weibull parameters, as well as water depths and distance to the nearest shore. The output of the tool is the average annual power production, power intensity (power per unit area), installation costs, operation and maintenance costs, and the levelized cost of energy.

In the developed model, we employ the atmospheric wind farm boundary layer model of Frandsen [17] as a backbone for assessing the wind turbine array (wake) effect in the atmospheric boundary layer, and combine it with a newly developed wind resource model based on the site Weibull distribution adjusted for wake effects. To our knowledge, such an approach has not been developed before, although this part of the model is important, as it allows us to determine the change of the Weibull parameters of the wind field within the wind farm due to wake effects. Furthermore, as the model of Frandsen [17] essentially is for an infinite wind farm, a simple correction is introduced that allows computations of the power production for wind farms of finite size. As cost model, we employ a set of simple parametrical expressions for the most essential wind farm cost elements, such as costs of wind turbines, support structures, cables and electrical substations, as well as OM. To take into account the influence of wake effects on OM expenses, a new expression has been developed that includes the spacing between the wind turbines as a parameter. This expression is further complemented with a new expression that relates the OM expenses to the distance from the coast. Since transparency and validation are cornerstones in the model, we show all details of the model and validate it on actual performance and cost data for a series of full-scale wind farms.

In the following, we present the various elements of the developed model complex and subsequently validate its suitability to predict wind power production and associated costs by comparing it to available production and cost data of existing well-documented wind farms. The paper is organized as follows. In Section 2, we introduce the theory for the developed models, which is divided into a model for the wind turbine, a model for the wind farm array effect, a model for the annual power production, and models for costs of wind turbines, support structures, cables and electrical substations, and of OM. The models in Section 3 are validated by comparing simulated results with actual data of the Lillgrund, the Rødsand, and the Horns Rev wind farms. In Section 4, we discuss the outcome of the validation, and in Section 5 we conclude and outline the main findings of the study. Since

the derivation of the equations for the annual power production of a wind turbine located in a wind farm is somewhat complex and nontrivial, the full derivation is for clarity given in Appendices A and B.

## 2. Model Description

In the following, we present the various elements of the model, which are divided into wind turbine modeling, wind farm and wake models, wind resource modeling, and cost modeling.

### 2.1. Wind Turbine Modeling

The power production of a solitary wind turbine $P = P(U)$ may, at a given mean wind speed $U$ below the rated one, be approximated by the following generic expression:

$$P(U) = \alpha U^3 + \beta \tag{1}$$

where the coefficients $\alpha$ and $\beta$ are determined as

$$\alpha = \frac{P_G}{U_r^3 - U_{in}^3}, \quad \beta = -\frac{P_G U_{in}^3}{U_r^3 - U_{in}^3} \tag{2}$$

with $P_G$ denoting the rated (installed) generator power, $U_{in}$ is the cut-in wind speed, and $U_r$ is the rated wind speed. This expression obviously allows for zero turbine production at the cut-in wind speed.

The thrust and power coefficient are defined as

$$C_T \equiv \frac{T}{\rho A_R U^2}, \quad C_P \equiv \frac{P}{\rho A_R U^3}, \tag{3}$$

where $T$ is the axial force, or thrust, acting on the rotor and $P$ is the power generated by the rotor, $\rho$ is the air density, and $A_R = \frac{\pi}{4}D^2$ the rotor area, with $D$ denoting the rotor diameter. We assume that the wind turbine operates at its optimum (rated) condition $C_P = C_{P,rated}$ at wind speeds lower than the rated wind speed $U_r$ and at a constant power yield $P = P_G$ at wind speeds higher than the rated wind speed. This operational strategy is typical for a modern wind turbine, which is operated with a variable tip speed at wind speeds below the rated one, and which is pitch-regulated at higher wind speeds. The rated wind speed is determined from Equation (3) in the condition where the generator operates at both maximum power and maximum (rated) power coefficient,

$$U_r = \sqrt[3]{\frac{8P_G}{\rho \pi D^2 C_{P,rated}}}. \tag{4}$$

With these assumptions, the wind turbine power curve is expressed as

$$P(U) = \begin{cases} \alpha U^3 + \beta; & U_{in} \leq U < U_r \\ P_G; & U_r \leq U \leq U_{out} \end{cases}. \tag{5}$$

where the wind turbine cut-out wind speed is denoted as $U_{out}$. In the subsequent model applications, it is assumed that $U_{in} = 3 \text{ m/s}$ and $U_{out} = 25 \text{ m/s}$. The corresponding thrust coefficient $C_T$ is approximated as

$$C_T = \begin{cases} C_{T,rated}; & U_{in} \leq U < U_r \\ C_{T,rated}(U_r/U)^{3/2}; & U_r \leq U \leq U_{out} \end{cases}. \tag{6}$$

In these expressions, it is implicitly assumed that the wind speed $U$ refers to the upstream wind speed at hub height of the wind turbine. If not otherwise stated, the values $C_{T,rated} = 0.75$ and $C_{P,rated} = 0.48$ are employed in what follows.

### 2.2. Wind Farm Modeling

To simplify the model, the only input regarding the wind farm topology is the total number of wind turbines $N_T$ and the wind farm area $A$. Denoting the assumed uniform distance between the wind turbines as $L_T$, and assuming that each wind turbine occupies a square of area $L_T^2$, the required wind farm area relates to the number of wind turbines and the spacing distance as

$$A = \left[ L_T \left( \sqrt{N_T} - 1 \right) \right]^2 = S^2 D^2 \left( \sqrt{N_T} - 1 \right)^2, \tag{7}$$

where $S = \frac{L_T}{D}$ is the wind turbine interspacing expressed in rotor diameters. This expression is obtained by assuming a quadratic wind farm topology with side length $L_T \left( \sqrt{N_T} - 1 \right)$. It is obvious that not all wind farms are quadratic, but this assumption constitutes a simple way of determining the average distance between the wind turbines within an acceptable accuracy independent of specific topology. In reality, the wind farm topology may take a multitude of forms that are not known in the initial planning phase of a wind farm. Rearranging Equation (7), we get

$$S = \frac{\sqrt{A}}{D \left( \sqrt{N_T} - 1 \right)}. \tag{8}$$

### 2.3. Wake Modeling

The wake model used to assess the wind power resource was originally developed by Templin [15] and later developed further by Frandsen [17,22]. This model assumes that the wind farm is so large that the wind field inside the wind farm is in equilibrium with the ambient atmospheric boundary layer (ABL) flow field. The model relies on the following assumptions [22]:

- The wind farm is large enough for the vertical wind profile to be horizontally homogeneous.
- The thrust on the wind turbine rotors is assumed concentrated at hub height.
- The horizontally homogeneous vertical wind profile is logarithmic both below and above hub height.
- The vertical wind profile is continuous at hub height.
- The height of the planetary boundary layer is considerably larger than the wind turbine hub height.
- Turbulent wind speed fluctuations are horizontally homogeneous.

It is assumed that the wind turbines create two logarithmic boundary layers, matched at hub height $h$ by the shear forces exerted by the wind turbines on the flow. Hence, the vertical profile of the horizontally averaged wind speeds takes the following form:

$$U_{lo}(z) = \frac{u_{lo}^*}{\kappa} \ln \left( \frac{z}{z_{0,lo}} \right) \; for \; z < h, \tag{9a}$$

$$U_{hi}(z) = \frac{u_{hi}^*}{\kappa} \ln \left( \frac{z}{z_{0,hi}} \right) \; for \; z > h, \tag{9b}$$

where $\kappa$ is the von Kármán constant ($\approx 0.41$), $u_{lo}^*$ and $u_{hi}^*$ are the friction velocities for the lower and upper boundary layers, respectively, and $z_{0,lo}$ and $z_{0,hi}$ are the corresponding roughness lengths. The friction velocity is per definition given as $u^* = \sqrt{\frac{\tau_W}{\rho}}$, where $\tau_W$ is the friction stress at the wall. With the two velocity profiles being continuous at hub height, we obtain the following relation:

$$U_{lo}(h) = U_{hi}(h) \Rightarrow u_{hi}^* \ln \left( \frac{h}{z_{0,hi}} \right) = u_{lo}^* \ln \left( \frac{h}{z_{0,lo}} \right). \tag{10}$$

From momentum conservation, the difference in shear stress above and below hub height is counteracted by the total thrust per area unit exerted on the flow. Since each wind turbine occupies an area, $L_T^2$, we get

$$\tau_{W,hi} - \tau_{W,lo} = \rho \frac{\pi}{4} D^2 \overline{U}_h^2 C_T / L_T^2 \Rightarrow u_{hi}^{*2} - u_{lo}^{*2} = c_t U_h^2, \tag{11}$$

where $c_t = \frac{\pi C_T}{8S^2}$, in which the thrust coefficient and dimensionless spacing between the wind farm turbines are determined from Equations (6) and (8), respectively. To close the system of equations, the following approximated formula for the geostrophic drag law is utilized [22],

$$u_{lo}^* = \frac{\kappa G}{\ln\left(\frac{G}{(fe^{A^*})z_{0,lo}}\right)}, \tag{12}$$

where $G$ is the geostrophic wind speed, $f = 2\Omega \sin \varphi$ is the Coriolis parameter, in which $\Omega$ denotes the rotational speed of the earth, and $\varphi$ is the latitude, and the constant $A^* = 4$ at latitude $55°$ (see [22]).

Combining Equations (9)–(12) results in the following simple equation to determine the mean wind speed at hub height inside the wind farm:

$$U_h = \frac{G}{1 + \ln\left(\frac{G}{f \cdot h}\right) \frac{\sqrt{c_t + \left(\kappa / \ln\left(h/z_{0,lo}\right)\right)^2}}{\kappa}}. \tag{13}$$

Input to the model is essentially the thrust coefficient of the wind turbines $C_T$, the dimensionless interspacing $S$ (which together determines the dimensionless wind farm parameter, $c_t$), plus the surface roughness of the sea surface $z_{0,lo}$, and the geostrophic wind speed $G$. In most cases, only the undisturbed mean wind speed $U_{h,0}$ at a given height $h$ is known for a specific site. The geostrophic wind speed is then determined indirectly by setting $c_t = 0$ and solving Equation (13) for $G$. For more details about the model, the reader is referred to [17,22].

### 2.4. Wind Resource Modeling

In order to determine the wind farm power production, and further to provide input to the applied cost model for wind farm OM expenses, we need to estimate the mean wind speed statistics both for the site without wind turbines (i.e., the ambient wind speed statistics) and for the internal wind farm flow field. Simple expressions for this are presented in the following section. For the detailed derivations, the reader is referred to Appendix A.

Ambient wind speed statistics over the year (typically based on 10 or 30 min averaging periods) are traditionally quantified using a two-parameter Weibull distribution. The probability density function (pfd) of a Weibull distributed random variable is

$$f(x; \lambda, k) = \begin{cases} \frac{k}{\lambda}\left(\frac{x}{\lambda}\right)^{k-1} e^{-\left(\frac{x}{\lambda}\right)^k}; & x \geq 0 \\ 0; & x < 0 \end{cases} \tag{14}$$

where $x$ is a realization of a stochastic variable $X$, $k > 0$ is the Weibull shape parameter, and $\lambda > 0$ is the Weibull scale parameter. The yearly average production of the wind turbine $P_y$ may be formulated as a convolution of the wind turbine production characteristics, Equation (1), with the mean wind speed probability density function expressed in Equation (14). Thus

$$P_y = \int_{U_{in}}^{U_{out}} P(U) f(U; \lambda, k) dU$$

$$= \alpha \int_{U_{in}}^{U_r} U^3 f(U; \lambda, k) dU + \beta \int_{U_{in}}^{U_r} f(U; \lambda, k) dU + P_G \int_{U_r}^{U_{out}} f(U; \lambda, k) dU \tag{15}$$

Carrying out the integration (see Appendix A), we obtain the following closed form expression for the average power production for a single standalone wind turbine:

$$P_y = \alpha\lambda^3 \left[ \Gamma\left( \frac{3+k}{k}, \left(\frac{U_{in}}{\lambda}\right)^k \right) - \Gamma\left( \frac{3+k}{k}, \left(\frac{U_r}{\lambda}\right)^k \right) \right] + \beta\left( e^{-\left(\frac{U_{in}}{\lambda}\right)^k} - e^{-\left(\frac{U_r}{\lambda}\right)^k} \right) + P_G\left( e^{-\left(\frac{U_r}{\lambda}\right)^k} - e^{-\left(\frac{U_{out}}{\lambda}\right)^k} \right) \quad (16)$$

where $\Gamma\,(*,*)$ is the incomplete gamma function (cf., Abramowitz and Stegun [23]), and $\alpha$ and $\beta$ are given from Equation (2).

The wind speed statistics inside a wind farm are different from the wind speed statistics of the ambient undisturbed flow. This is due to the wind speed reduction caused by the wake effects from neighboring wind turbines, which in the present model is determined from Equation (13). To determine the average power production within the wind farm, the Weibull scale parameter $\lambda$ needs to be adjusted to account for the wind flow of the turbines operating in the wind farm. The full derivation of the equations forming the power production of a wind turbine located inside a wind farm is shown in Appendix A. Here we only show the final result, which is given by the following closed form expression:

$$
\begin{aligned}
P_{WF,y} &= \alpha(\varepsilon_1\lambda)^3 \left[ \Gamma\left( \frac{3+k}{k}, \left(\frac{U_{in}}{\varepsilon_1\lambda}\right)^k \right) - \Gamma\left( \frac{3+k}{k}, \left(\frac{U_r}{\varepsilon_1\lambda}\right)^k \right) \right] + \beta\left( e^{-\left(\frac{U_{in}}{\varepsilon_1\lambda}\right)^k} - e^{-\left(\frac{U_r}{\varepsilon_1\lambda}\right)^k} \right) \\
&+ P_G\left( e^{-\left(\frac{U_r}{\varepsilon_2(U_r)\lambda}\right)^k} - e^{-\left(\frac{U_{out}}{\varepsilon_2(U_{out})\lambda}\right)^k} \right)
\end{aligned}
\quad (17)
$$

where the following abbreviations are introduced,

$$\varepsilon_1 = \frac{1 + \frac{\gamma}{\delta}}{1 + \frac{\gamma}{\kappa}\sqrt{\frac{\pi C_{T,rated}}{8S^2} + (\kappa/\delta)^2}}, \quad \varepsilon_2 = \frac{1 + \frac{\gamma}{\delta}}{1 + \frac{\gamma}{\kappa}\sqrt{\frac{\pi C_{T,rated}}{8S^2}(U_r/U_h)^{3/2} + (\kappa/\delta)^2}} \quad (18)$$

And

$$\gamma = \ln\left( \frac{G}{fh} \right), \quad \delta = \ln\left( \frac{h}{z_{0,lo}} \right). \quad (19)$$

### 2.5. Finite-Size Wind Farm Correction

Since the performance model introduced in Sections 2.1–2.4 essentially is based on a fully developed atmospheric boundary layer in equilibrium with an infinite wind farm, it is necessary to introduce a correction for the finiteness of a real wind farm. It is expected that such a correction will depend on the number of wind turbines along the edges of the farm relative to the total number of wind farm turbines, implying that this correction approach asymptotically vanishes when the wind farm size approach "infinity". Since the wind turbines can be placed in a multitude of different geometrical topologies, as a general expression, we use the number of wind turbines along each edge of the chosen generic topology $\sqrt{N_T}$, which in turn corresponds to a total of $4\sqrt{N_T}$ edge turbines. Since the wind turbines along outer edges are subject to the free wind, the energy production from about half of these (assuming a uniform wind direction distribution) will, in average, be equal to that of a standalone wind turbine. Furthermore, depending on the wind direction, some of the wind turbines inside the farm that are located close to the edge will also be exposed to ambient wind conditions (see Figure 1). Therefore, part of the wind turbines will be in fully developed wake condition and some will be partly exposed to wake effects, whereas those along the edges pointing towards the wind direction will not be exposed to wake effects at all.

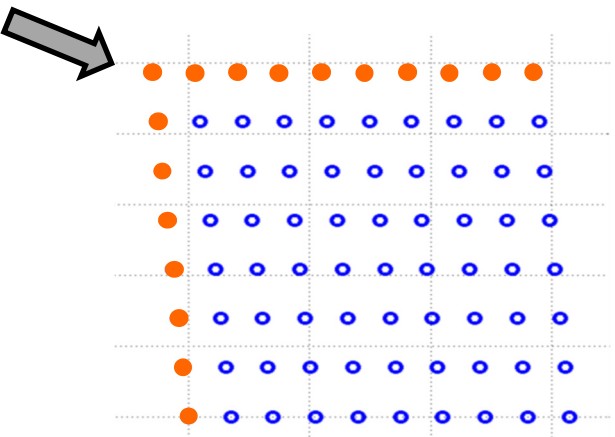

**Figure 1.** Sketch of a wind farm (Horns Rev 1) with wind vector from northwest, where the wind turbines along the first northern and the first western line (red dots) are seen to be directly exposed to the free wind.

As a simple heuristic approximate expression for taking into account the finiteness of the farm, we use the following expression for the wind farm power production:

$$P_E = (N_T - a\sqrt{N_T})P_{WF,y} + a\sqrt{N_T}P_y, \tag{20}$$

where $a$ is a correction constant that ideally should be calibrated against performance data from actual wind farms or computational high-fidelity results from, e.g., Navier–Stokes simulations. In the following, we employ the value $a = 3$, stating that the average number of wind turbines subject to freestream conditions corresponds to three-quarters of the turbines located along the edge of the wind farm.

*2.6. Cost Models*

Cost models are needed for any economic optimization aimed at finding the optimal balance between wind farm production, operational costs, and financial costs. Given the broad and generic character of the present study, relatively simple models must be used. These cost models, as well as the assumptions on which they are based, are described in the following sections.

2.6.1. Cost of Wind Turbine

The cost of a wind turbine in MEUR, $C_{WT}$, may according to Lundberg [6] be expressed as $C_{WT} = -0.15 + 0.92P_G$, where $P_G$ is the installed generator power in MW. However, this pricing refers to the year 2003, when the report was compiled. The inflationary development in (Danish) consumer prices in general from the year 2003 up to 2015 was 23% [5,24]. In this study, we will assume wind turbine prices follow the inflation in general consumer prices during this period, and we will further add 2% to approximately include the wind turbine price development up to today (i.e., 2019). With these assumptions, we finally arrive at the following expression for wind turbine prices in MEUR:

$$C_{WT} = 1.25 \cdot (-0.15 + 0.92P_G). \tag{21}$$

2.6.2. Cost of Support Structure

Cost and type of wind turbine support structures depend primarily on wind turbine size and water depth. A monopole foundation is considered advantageous for shallow water regimes, which in the present context means water depths down to about 35 m. For water depths beyond 35 m, jacket foundations are convenient and consequently assumed.

The cost of a monopile support structure in MEUR, $C_{FM}$, may in a first order approximation be simplified as (Buhl and Natarajan [25])

$$C_{FM} = \frac{P_G(H^2 + 100H + 1500)}{7500}, \tag{22}$$

where $P_R$ denotes the wind turbine rated power in MW, and $H$ is the water depth in meters.

Cost of a jacket support structure in MEUR, $C_{FJ}$, may in a first order approximation be simplified as (Buhl and Natarajan [25])

$$C_{FJ} = \frac{P_G(0.5H^2 - 35H + 2500)}{7500}. \tag{23}$$

### 2.6.3. Cost of Wind Farm Electrical Grid

Assuming the internal electrical grid predominantly (i.e., except for one connecting line along the alternative direction) laid out along one of the directions in the quadratic grid, the aggregated length of the grid cables $L_C$ is given by

$$L_C = SD\left(\sqrt{N_T}+1\right)\left(\sqrt{N_T}-1\right) = SD(N_T - 1). \tag{24}$$

The wind farm grid financial costs per running meter, including cable cost and costs of installation, for an offshore site is taken as $C_C = 675$ EUR (Rethoré et al. [26] and Larsen et al. [27]). Consequently, the total aggregated grid costs $C_G$ are given as

$$C_G = L_C C_C. \tag{25}$$

### 2.6.4. Cost of Operation and Maintenance

The cost of OM depends on wind turbine size as well as on wind turbine spacing, in the sense that a smaller spacing, and thereby higher loadings, increases the costs and, for larger wind turbines, these costs are reduced per installed MW. It is reasonable to assume that the relative wind turbine size effect (e.g., the relative reduction in OM for one 6 MW wind turbine compared to two 3 MW wind turbines) for wind turbines subjected to identical load conditions is independent of the particular load level, and we will consequently assume that the size and load dependencies can be factorized as

$$C_{O\&M,base}(P_G, S) = f_{WT}\left(P_G\big|P_{G,\,Ref}\right) \cdot C_{WT_{Ref}} \cdot f_C \cdot f_S(S), \tag{26}$$

where $f_{WT}\left(P_G\big|P_{G,Ref}\right)$ is the wind turbine size factor, $C_{WT_{Ref}}$ is the yearly cost of OM for a reference wind turbine with rated power $P_{G,\,Ref}$ operating under ideal conditions with a wind turbine capacity factor equal to one, $f_C$ is the wind turbine capacity factor for an imaginary solitary wind turbine at the site of interest, and $f_S(S)$ is a load factor accounting for the impact of the wind farm load level—and thus of the wind turbine spacing—on the OM costs. The load factor depends on the load condition for the particular wind farm turbine, and it is expressed in terms of wind farm topology (i.e., spacing) as

$$f_S(S) = \frac{P_{S,y}}{P_{WF,y}} = \left(\frac{P_{S,y}}{P_G}\right) \Big/ \left(\frac{P_{WF,y}}{P_G}\right) = \frac{f_C}{f_{WF}}, \tag{27}$$

where $P_{S,y}$ is the average annual power yield of a solitary wind turbine at the site of interest, $P_{WF,y}$ is the average annual power yield of a wind farm turbine, and $f_{WF} = P_{WF,y}/P_G$ is the wind farm capacity factor. As seen, the load factor increases for decreasing wind farm capacity factor (and vice versa), reflecting increased wake impact and thus in turn increased loading.

Inspired by Berger [28], where a 14% reduction of annual OPEX cost per MW is stated by shifting from 3 MW to 6 MW wind turbines, we will assume that this relative reduction

can be linearly extrapolated to other wind turbine sizes within a wind turbine size regime spanned by half and double the size of the reference wind turbine, respectively. Outside this wind turbine size regime, it seems reasonable to assume an exponential behavior, where 14% reduction of OPEX is gained for a doubling of the wind turbine size, and a corresponding increase of OPEX results if the wind turbine size is halved. Thus, for an increase in wind turbine size, the wind turbine size factor is quantified as

$$f_{WT}\left(P_G \middle| P_{G,\,Ref}\right) = \begin{cases} 1 - \dfrac{0.14\left(P_G - P_{G,Ref}\right)}{P_{G,\,Ref}} \text{ for } P_{G,Ref} \leq P_G \leq 2P_{G,Ref} \\ 0.86^{0.5\,P_G/P_{G,Ref}} \text{ for } 2_{G,\,Ref} < P_G \end{cases}. \qquad (28)$$

For a decrease in wind turbine size the analog expression is

$$f_{WT}\left(P_G \middle| P_{G,\,Ref}\right) = \begin{cases} 1 - \dfrac{0.325\left(P_G - P_{G,Ref}\right)}{P_{G,Ref}} \text{ for } 0.5P_{G,Ref} \leq P_G \leq P_{G,Ref} \\ 0.86^{-0.5P_{G,Ref}/P_G} \text{ for } P_G < 0.5P_{G,Ref} \end{cases}. \qquad (29)$$

Note that the difference in factors in the linear expressions relates to the reference wind turbine being the smallest and the largest wind turbine in these expressions, respectively. The reference wind turbine is for the present study taken as a 10 MW wind turbine, for which the OM costs per year may be specified as $C_{WT,Ref}$ = 106 EUR/kW (Chaviaropoulos and Natarajan [29]). This number, which forms the basis for the OM model in Equation (26), is determined as an average value based on analyses of different offshore located wind farms. However, the distance to the shore is an additional important parameter, as the OM expenses depend on the transport time, and hence the distance from the shore to the location of the wind farm. Therefore, the cost model formulated in Equations (26)–(29) forms the basis for OM expenses at a certain average location in terms of distance to service harbor. To determine the actual OM expenses, Equation (26) needs to be corrected by an additional term, which depends on the distance to the shore:

$$C_{O\&M}(P_R, S, Y) = C_{O\&M,base}(P_R, S) + C_{O\&M,L}(Y), \qquad (30)$$

where $Y$ designates the distance to the shore. In a first order approximation, a simple way of determining the distance-dependent parameter is to assume a linear relationship, i.e., reflecting proportionality in transportation time and vessel fuel consumption:

$$C_{O\&M,L}(Y) = \frac{\Delta C_{O\&M}}{\Delta Y}\left(Y - Y_{ref}\right), \qquad (31)$$

where $Y_{ref}$ is a reference distance referring to the value in Equation (31). Since this number mainly is based on experience from the Rødsand and Horns Rev wind farms, we assume a reference distance corresponding to the average shore distance of these wind farms. As a rough and ready to use number, we therefore define $Y_{ref}$ = 20 km. The gradient $\frac{\Delta C_{O\&M}}{\Delta Y}$ needs to be determined from experience. One way to do this is to consider the difference in agreed cost price between two nearby located wind farms. Assuming that the invested installation expenses per produced energy unit is the same for the two wind farms, then this difference approximately corresponds to the difference in OM expenses. Using the Horns Rev I and Horns Rev II wind farms, which are located 14 km from each other, and which operate at agreed cost prices of 0.432 DKK/kWh and 0.518 DKK/kWh ([30,31]), respectively, we obtain the following estimate:

$$\frac{\Delta C_{O\&M}}{\Delta Y} = \frac{0.518 \text{ DKK/kWh} - 0.432 \text{ DKK/kWh}}{14 \text{ km}} = 0.0061 \text{DKK/kWh·km} = 0.82 \cdot 10^{-3} \text{ €/kWh·km}$$

where the exchange rate 1 EUR = 7.50 DKK has been used. Assuming an average profit rate of 13% ([32]), then 87% of the result in the above estimate is expected to cover the difference in OM expenses. If we further introduce the wind farm capacity factor $f_{WF}$, we get the following expression for the dependence of OM costs on the distance to the shore:

$$C_{O\&M,L}(Y) = 0.87 \cdot 0.82 \cdot 10^{-3} \cdot 8760 \cdot f_{WF}\left(Y - Y_{ref}\right) = 6.24 f_{WF}\left(Y - Y_{ref}\right), \tag{32}$$

where $8760 \cdot f_{WF}$ is the number of full load hours within a year. This means that the unit is EUR/kW, with $\left(Y - Y_{ref}\right)$ measured in kilometers. Since Equation (32) is linear and based on only two "observations", it is a priori not expected to be valid for longer distances from the coast than those analyzed. Therefore, in the present work, we only use it for wind farms located up to 35 km from the coast, corresponding to, e.g., the Horns Rev 3 wind farm.

Because OM costs are running costs, contrary to the financial costs described in Sections 2.6.1–2.6.3, which refer to the time of the wind farm installation, we need assumptions on the development of OM costs over time in comparison with the inflation. We will here assume that the development of OM costs over time follows the inflation in general. This makes the rate of inflation the natural choice for the discounting rate, and with this choice we conveniently avoid computation of net present values by letting all prices refer to the time of wind farm installation.

### 2.6.5. Levelized Cost of Energy

Costs other than those described in the previous sections—e.g., cost of transformer station(s) and establishment of a main cable to the coast—are presumed to depend only on the rated production of the wind farm and thus for the present study independent of the wind farm layout (i.e., wind turbine spacing) and the choice of wind turbine size. Such costs will affect the levelized cost of energy estimation, and to arrive at reasonable realistic LCOE estimates we will, in line with Mahulja [33], assume that costs of wind turbines, internal wind farm grid, and foundations account for 75% of the total investment cost, which is based on experiences from the Danish Horns Rev and Nysted offshore wind farms (Morthorst and Kitzing [4]). The remaining 25% is mainly due to electrical infrastructures, such as onshore cables and substations. However, this is an average number for wind farms located at different distances from the coast, and for an actual wind farm, it obviously increases as function of distance to the shoreline. To estimate this dependency, we employ data from the ENS-report [34], which provides the following relative costs: (1) 10% for planning, development, and financing; (2) 9% for the substation; and (3) 3% for export cable; therefore, 25% in total. Of these, only the export cable cost is a variable cost that depends on distance to the shore. In rough numbers, the export cable constitutes one-quarter of the 25% "additional costs", $C_{add}$. Assuming that the above stated cost of the export cable relates to an average distance $Y_{ref}$ of approximately 20 km from the coast, then splitting into fixed and variable costs results in the following cost estimate:

$$C_{add}(Y) = C_p + C_s + \frac{Y}{Y_{ref}} C_{Y_{ref}}, \tag{33}$$

where $C_p$ is the planning cost, $C_s$ is the substation cost, and $C_{Y_{ref}}$ is the cost of a 20 km export cable. Normalizing the above cost estimate with the total cost $C_{total}$ of an offshore wind farm installation, we find

$$\frac{C_{add}}{C_{total}} = 0.19 + \frac{Y}{Y_{ref}} 0.06. \tag{34}$$

The total costs, often referred to as CAPEX, of the wind farm amount to

$$C_{total} = C_{add} + N_T \left[ C_{WT} + \gamma_F C_{FM} + (1 - \gamma_F) C_{FJ} \right] + C_G, \tag{35}$$

where $\gamma_F$ is the fraction of wind turbines erected on monopole foundations, and $(1 - \gamma_F)$ is the fraction of wind turbines erected on jacket foundations. The total installation cost is consequently given as

$$C_{total} = \frac{\left[ N_T \left[ C_{WT} + \gamma_F C_{FM} + (1 - \gamma_F) C_{FJ} \right] + C_G \right]}{\left[ 0.81 - 0.06 \frac{Y}{Y_{ref}} \right]}. \tag{36}$$

The estimated LCOE, expressed in terms of a kWh price and defined as capital expenditure plus OM costs divided by the total production, is given by

$$LCoE = \frac{\left[ N_T \left[ C_{WT} + \gamma_F C_{FM} + (1 - \gamma_F) C_{FJ} \right] + C_G \right] / \left[ 0.81 - 0.06 \frac{Y}{Y_{ref}} \right] + N_T N_Y P_G C_{O\&M}}{8.76 N_Y P_E}, \tag{37}$$

where $N_Y$ is the life time of the wind farm in years, $P_E$ is the yearly average power production for the wind farm (Equation (20)), and the denominator is the total electricity production in kWh. For the present study, we assume a wind farm life time of 20 years, i.e., $N_Y = 20$.

## 3. Results

To validate the combined energy production and the suite of cost models, we consider a series of wind farms of different sizes and locations from Danish shores. In the following, we give the details of the wind farms and compare actual data for energy production and costs with estimates resulting from the model.

### 3.1. Wind Farm Cases

The wind farms used for the validation study are the Lillgrund wind farm (LG), located in Øresund between Denmark and Sweden, and Rødsand 1 (RS1), Rødsand 2 (RS2), Horns Rev 1 (HR1), Horns Rev 2 (HR2), and Horns Rev 3 (HR3), which all are located in Danish waters. As will be clear from the following description, these wind farms are different in terms of size, location, and geometry, and together they cover nearly two decades of experience with offshore located wind farms. To refer all costs to 2019 prices, we assume an average annual inflation of 2% in the period 2002 to 2016 (see Gonzales-Rodriguez [5]) and a corresponding value of 1% from 2016 to 2019 [35]. Most production data are taken from Andrew [36], which contains updated data from all large Danish offshore wind farms.

- Lillgrund Wind Farm (LG)

The Lillgrund wind farm, owned by Vattenfall AB, is located about 10 km from the coast of South Sweden (Skåne) on water depths ranging from 4 to 8 m. The park consist of 48 Siemens SWT 2.3-93 wind turbines and, according to [37], it produces about 330 GWh per year. The wind farm has been fully operational since June 2008, at which time the cost of the wind farm was 1.8 billion SEK, corresponding approximately to 180 MEUR. With an accumulated inflation rate of 19%, we obtain a cost price of 214 MEUR in 2019 prices.

- Rødsand 1 (RS1)

Rødsand 1, also referred to as Nysted, is owned by the Danish utility company Ørsted in a consortium together with PensionDanmark and Stadtwerke Lübeck. The wind farm is located 13 km south of Nysted in a shallow water area with water depths ranging between 6 and 10 m. The park consists of 72 Siemens SWT 2.3-82 wind turbines and produces about 570 GWh per year [30,36]. The wind farm was commissioned in 2003, and except for a 4.5 month stop in 2007, it has been fully operational ever since. According to Gerdes et al. [38] the installation costs amounted to 250 MEUR in 2003 prices, corresponding to

322 MEUR in 2019 prices. It was warranted an electricity price of 0.45 DKK per kWh for the first 42,000 full-load hours, and is presently subject to a power purchase agreement of 0.629 DKK per kWh.

- Rødsand 2 (RS2)

Rødsand 2, which is located 3 km west of Rødsand 1, was commissioned by E.ON in 2010, but since 2013 operated by SEAS-NV. The park consists of 90 Siemens SWT 2.3-93 wind turbines, which produce about 830 GWh per year [36]. According to [39], the installation costs amounted to 384 MEUR in 2010, corresponding to 460 MEUR in 2019 prices. After some adjustments, it is, like Rødsand 1, now subject to a power purchase agreement of 0.629 DKK per kWh.

- Horns Rev 1 (HR1)

Horns Rev 1 (see Figure 2) was the first offshore wind farm in the North Sea. It was built by the Danish energy company Elsam (now part of Ørsted). It is located 14–20 km west of Jutland on 6–14 m water depth. The farm consists of a total of 80 Vestas V80-2.0 MW units, which were installed by the Danish offshore wind farms services provider A2SEA, with the last wind turbine coming into operation in December 2002. It received a guaranteed price of 0.453 DKK/kWh for the first 42,000 h, which was later negotiated to 0.432 DKK/kWh [31]. Since 2005, the wind farm has been owned by Ørsted (40%) and Vattenfall (60%), see [30]. The installation price in 2002 amounted to 270 MEUR, corresponding to 354 MEUR in 2019 prices, and the power production is approximately 580 GWh per year [36].

- Horns Rev 2 (HR2)

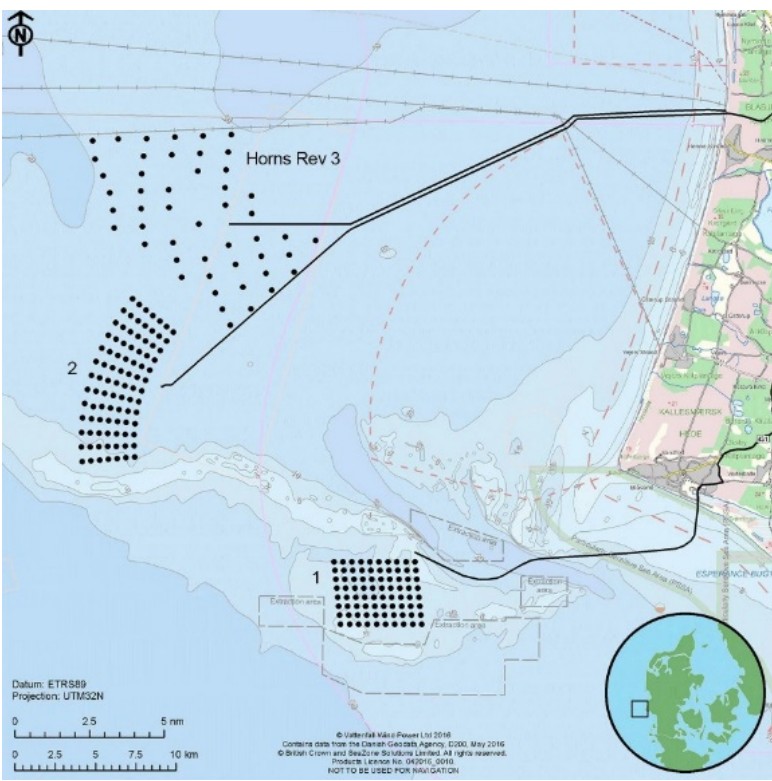

**Figure 2.** Sketch of the three Horns Rev wind farms (published by permission of Vattenfall Wind Power). To the bottom left, the scale indicates the size of the wind farms and the map at the bottom right shows the position on a map of Denmark. It is seen that the layout of the oldest wind farm (Horns Rev 1) has a nearly quadratic shape, whereas Horns Rev 2 is curved, and the wind turbines forming Horns Rev 3 are distributed according to an optimization algorithm, which locates them in a seemingly less systematic manner.

The Horns Rev 2 wind farm (see Figure 2), owned by Ørsted, was completed and commissioned in September 2009. It is located approximately 30 km west of Jutland on 9–17 m water depth, and consists of 91 Siemens SWT 2.3-93 units. It received a guaranteed price of 0.518 DKKkWh for the first 50,000 h. The installation price in 2009 amounted to 450 MEUR, equivalent to 524 MEUR in 2019 prices. According to [30], the power production is estimated to approximately 800 GWh per year, but seems to be about 10% higher, reaching a value of 880 GWh/year [26].

- Horns Rev 3 (HR3)

Horns Rev 3 (see Figure 2) is the newest of the Horns Rev wind farms. It was built by Vattenfall and decommissioned in August 2019 [40]. It is located 25–40 km west of Jutland on 11–19 m water depth, and consists of 49 MHI Vestas V164-8.0 wind turbines. At the moment, the wind farm receives a guaranteed price of 0.77 DKK/kWh, which is expected to be lowered to 0.59 DKK/kWh [41]. The installation price was approximately 1000 MEUR in 2019 prices. According to Vattenfall [40], the power production is anticipated to be approximately 1700 GWh per year.

The data, which form the input to the numerical model, are summarized in Table 1 for the various wind farms. Here, $P_g$ is the nameplate generator capacity, $D$ is the rotor diameter, $H_t$ is the tower height, $D_w$ is the water depth, $L_s$ is the average distance to the shore, $A$ is the area of the wind farm, $N_t$ denotes the number of wind turbines, and $\lambda$ and $k$ are the two Weibull parameters. Furthermore, for comparison and validation, installation costs, CAPEX, annual energy production, $E$, and power purchase agreement, PPA, are given in the last three columns of the table. The units of the different parameters are also indicated in the table. Note, that PPA, which is only given for Danish wind farms, is originally given in Danish currency (DKK) per KWh. In the table, these are transferred to EUR per MWh (1 EUR = 7.50 DKK), and in the following all costs are given in 2019 prices. Weibull parameters are taken from [42]. It should be emphasized that the Weibull parameters are used to determine the mean wind speed and the geostrophic wind speed, cf., Equation (13).

**Table 1.** Wind farm data used for validation of the model.

|  | Pg [MW] | D [m] | Ht [m] | Dw [m] | Ls [km] | A [km²] | Nt | λ [m/s] | k | CAPEX [MEUR] | E [GWh] | PPA [EUR/MWh] |
|---|---|---|---|---|---|---|---|---|---|---|---|---|
| **LG** | 2.3 | 93 | 65 | 4–8 | 10 | 4.8 | 48 | 9.7 | 2.4 | 214 | 330 | N/A |
| **RS1** | 2.3 | 82 | 69 | 6–10 | 13 | 22 | 72 | 10.5 | 2.4 | 322 | 540 | 83.9 |
| **RS2** | 2.3 | 93 | 68 | 6–10 | 16 | 35 | 90 | 10.5 | 2.4 | 460 | 790 | 83.9 |
| **HR1** | 2.0 | 80 | 70 | 6–14 | 16 | 20 | 80 | 11.0 | 2.4 | 354 | 580 | 57.6 |
| **HR2** | 2.3 | 93 | 68 | 9–17 | 30 | 33 | 91 | 11.2 | 2.4 | 524 | 880 | 69.0 |
| **HR3** | 8.0 | 164 | 105 | 11–19 | 35 | 88 | 49 | 11.5 | 2.4 | 1000 | 1700 | 78.7 |

### 3.2. Computed Results

Using the geometrical and meteorological data shown in Table 1 as input, the model introduced in Section 2 is employed to compute the various parameters shown in Table 2. In the table, $S$ denotes the average distance (in rotor diameters) between the wind turbines, CF is the capacity factor, PoA is the power density, OPEX is the operational expenditures, given per produced energy-unit and as a total over 20 years, and LCOE is the levelized cost of energy.

**Table 2.** Computed wind farm results.

|  | S [-] | CF [%] | PoA [MW/km$^2$] | OPEX [EUR/MWh] | OPEX [MEUR] | CAPEX [MEUR] | E [GWh] | LCOE [EUR/MWh] |
|---|---|---|---|---|---|---|---|---|
| **LG** | 3.98 | 30.9 | 7.13 | 97.6 | 580 | 206 | 299 | 132 |
| **R1** | 7.64 | 39.0 | 2.94 | 32.7 | 371 | 336 | 566 | 62.4 |
| **R2** | 7.50 | 43.6 | 2.58 | 32.5 | 514 | 431 | 791 | 59.7 |
| **HR1** | 7.04 | 42.7 | 3.41 | 36.9 | 441 | 334 | 598 | 64.8 |
| **HR2** | 7.23 | 47.6 | 3.02 | 40.7 | 710 | 482 | 872 | 68.4 |
| **HR3** | 9.53 | 54.0 | 2.41 | 29.7 | 1100 | 937 | 1855 | 54.9 |

## 4. Discussion

From Table 2, it is seen that there is some spreading in the average wind turbine interspacing for the different wind farms, where Lillgrund has an average wind turbine distance that is less than 4 D, and Horns Rev 3 has a distance of close to 10 D between the wind turbines. The remaining parks are mutually more similar with average distances ranging between 7 and 8 D. This can also be seen in the capacity factor, which, due to wake effects, is smallest for Lillgrund and largest for Horns Rev 3. This tendency is also reflected in the cost of energy, where the LCOE of the Lillgrund site is more than twice as large as for the remaining sites. Thus, the wind farms are in many ways so different that they in total represent a challenging validation case for a combined resource assessment and cost model. In the following, the accuracy and generality of the developed model complex will be validated through a systematic comparison between computed and actual realized data.

In Figure 3, we compare computed and actual production data for the different wind farms. It is seen that, in spite of its simplicity, the developed model predicts an annual energy production, which is very similar to the actual production for the different wind farms. It should be mentioned that no actual production data yet exist for Horn Rev 3, and that the "observed" data in this case is based on the production estimated by the developer. It can also be seen that only the Lillgrund wind farm is under-predicted, which most likely is because the assumption of momentum equilibrium between the atmospheric boundary layer is challenged for such a small wind farm. Disregarding the Lillgrund and Horns Rev 3 wind farms, the computed production of the remaining wind farms is in excellent agreement with actual production data. For Horns Rev 2, it can be seen that the prediction from the presented minimalistic framework comes closer to the full-scale observations than the original predictions presented in [30], where the annual expected production when establishing the wind farm was estimated to be 800 GWh. This difference between the initial predictions by the developer, prior to the establishment of the wind farm, and the actual energy production gives an estimate of the uncertainties when planning wind farms, which here amounts to 10%. As can be seen from the comparisons in Figure 3, in spite of the simplicity of the present model, it is typically capable of predicting annual energy productions within 5% accuracy, as compared to actual production data.

The model's prediction ability can further be assessed by comparing its accuracy to the accuracy of other models. Today, there exists a range of analytical wake models in use by companies and developers of wind farms. In collaborative work between a series of wind farm developers and consultancy companies, different wake models were compared in Walker et al. [43]. As a result of the comparison, losses in annual production due wake effects were found to be between 15% and 35%, depending on the actual wind farm, and the accuracy for determining wake losses was about 25%. Hence, the overall accuracy of the wake models to determine the average annual power production is estimated to be in the range of 4–10%, which fits very well with the capability of the present model. However, it should be emphasized that the tested wake models demanded detailed knowledge about the layout of the tested wind farms and the actual wind turbines, as well as detailed wind

statistics regarding wind direction data. Therefore, we generally expect the "simplistic" model to be less accurate than the more detailed and more computational demanding models. In a recent study by Andersen et al. [44], a series of high-fidelity large eddy simulations (LES) were carried out to analyze the performance of large wind farms. As a part of the study, results from a couple of simple wake models, including the present one, were compared to the LES simulations. By including the surface roughness length as an additional parameter to model the turbulence intensity, the present model was found to perform very well, and it captures the power performance at both high and low turbulence intensity levels, as well as the gradual change in performance at all turbine spacings. Hence, a future development of the model could be to include the surface roughness as an active parameter in the model.

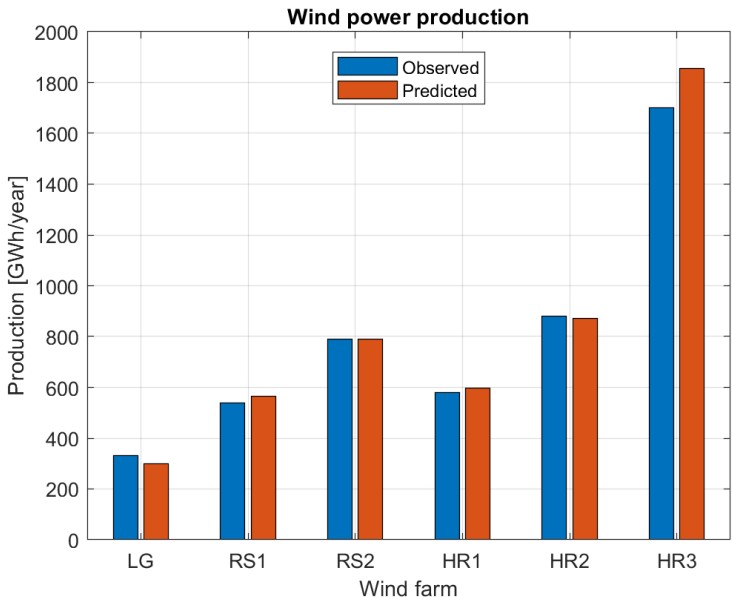

**Figure 3.** Comparison between computed and actual production data for the different wind farms.

Another way of presenting the wind farm production is to compute the average power performance per area unit (PoA). This is accomplished by taking the values shown in Figure 3 and dividing them with the number of hours per year (8760) and the area of the specific wind farm. The outcome is shown in Figure 4, where PoA is measured in MW/km$^2$ (or equivalently W/m$^2$). Comparing the power intensity, Lillgrund is, due its compactness, noticeably different from the rest, with 7.85 MW/km$^2$, whereas the remaining wind farms have a power intensity in the range from 2.2 to 3.3 MW/km$^2$. The relative difference between actual and predicted values are obviously identical to the values in Figure 3. Although the Lillgrund wind farm has very high power intensity, the costs are also high, with a LCOE that is more than double that for the other wind farms. The reason is that the wake loses are quite high, and that OPEX increases inversely with the distance between the turbines (Equations (26) and (27)). In fact, there will always be a tradeoff between erecting wind turbines close to each in order to maximize the power harvest per square unit, and to minimize wake losses by erecting the wind turbines far from each other. An example of the latter is the Horns Rev 3 wind farm, where the wind turbines in average are located about 9.5 diameters from each other, and the expected and predicted power intensity is 2.2 MW/km$^2$ and 2.4 MW/km$^2$, respectively, but the predicted energy costs only amounts to 54.9 EUR/MWh.

In Figure 5, we compare the capital expenses resulting from the model simulations and the realized actual data. A good agreement between the computed and the actual data is observed. The biggest discrepancy is seen for Horns Rev 2, where the model under-predicts CAPEX with about 10%. The general tendency, however, is that the predicted values compare with the actual costs within an error of 5%. It should also be mentioned,

that CAPEX is a relatively reliable parameter, as the main elements forming it are easily found in the open literature and that, due to legislation, the overall costs of establishing wind farms are publically available. This is in contrast to OPEX that are formed by many different elements, such as local weather conditions, logistics related to access to ports, helicopters, and vessels [45], as well as relatively unknown factors owing from the impact of sea conditions on the structural elements of the wind turbines. As an example of the latter, Horns Rev 1 constitutes an example, where additional expenses have been spent on exchanging gear boxes and turbine blades, and where the harsh weather conditions in the North Sea have resulted in different unforeseen maintenance problems, such as leading edge erosion on the blades [46].

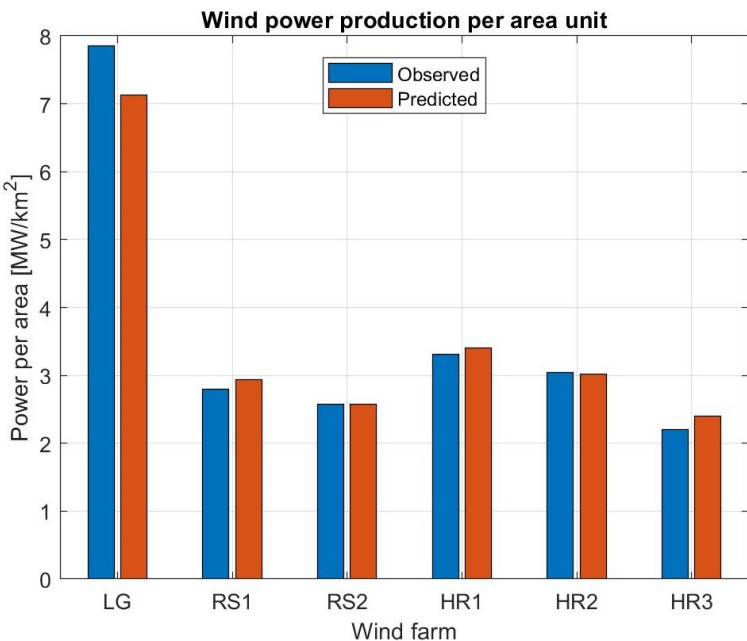

**Figure 4.** Comparison between computed and actual power per area unit for the investigated wind farms.

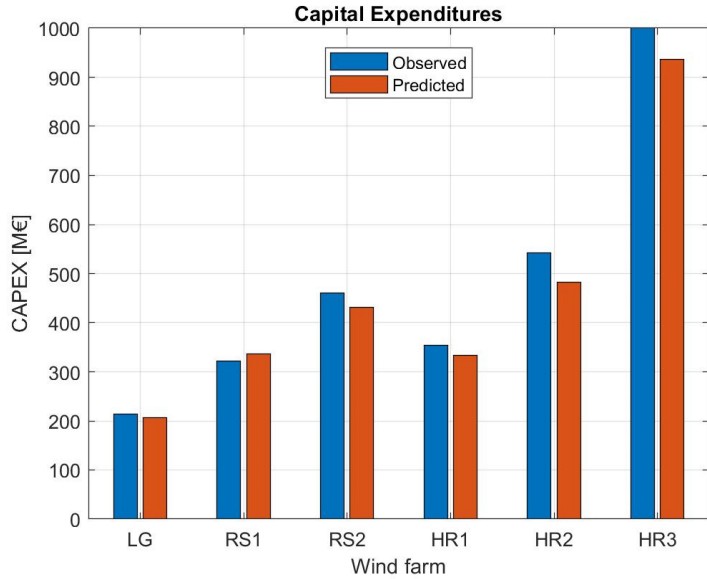

**Figure 5.** Comparison between computed and actual CAPEX for the different wind farms.

The levelized cost of energy is in Figure 6 compared to the minimum price guaranteed by the Danish government, in the figure referred to as PPA (power purchase agreement), for the different wind farms. Since the actual OM expenses (OPEX) are not publicly available information, we here employ values that are computed by our model. Hence, what is referred to as actual data in the figure is determined from actual costs and actual production data, however, with the OM expenses determined from the model. In principle, the difference between PPA and LCOE is the profit of producing the electricity. It is therefore a priori expected that PPA is larger than LCOE. As seen from the comparison, this is not always the case. Since the computed LCOE values for a specific wind farm in general deviate less than 10%, it is most likely the computed OPEX values that constitute the main uncertainty associated with determination the LCOE. Furthermore, the PPA may contain some specific agreements between the authorities and the utility company, which are not included in our cost model, like special agreements for overproduction, etc.

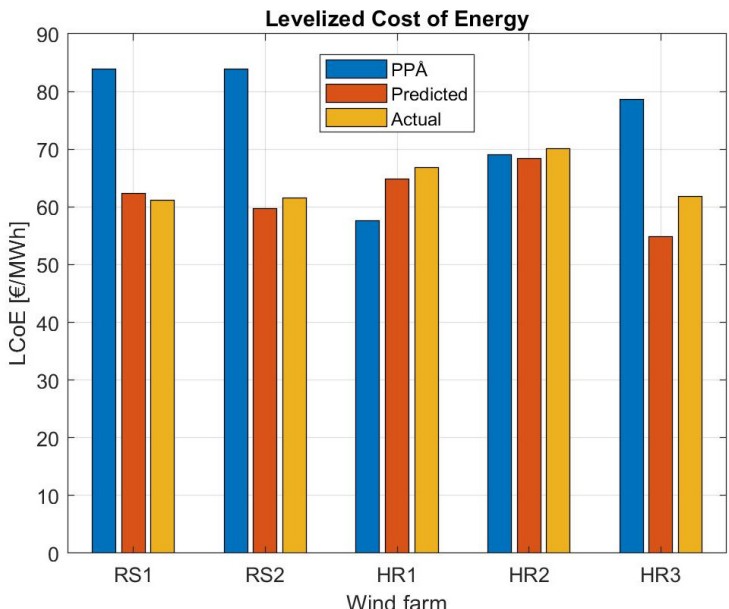

**Figure 6.** Comparison between agreed power purchase costs (PPA) and computed and actual levelized costs of energy for the analyzed wind farms. Since actual OM costs are not publically available, "actual" refers to actual CAPEX and actual production data, but with computed OPEX values.

In Figure 7, we determine and compare CAPEX per produced energy unit (EUR/MWh). The reason why we find this number interesting is that it seems to be invariant to the actual size and location of the wind farm. It is interesting that both actual and computed values are relatively similar, and further that the values are nearly identical for the various wind farms. As a very good estimate, a unified value of 30 EUR/MWh may be used as a rough and ready number. In fact, this property was one of the assumptions behind the derivation of the expression for the dependency of the OM expenses on the distance to the shore, Equation (32), and the present findings thus consolidate this approach.

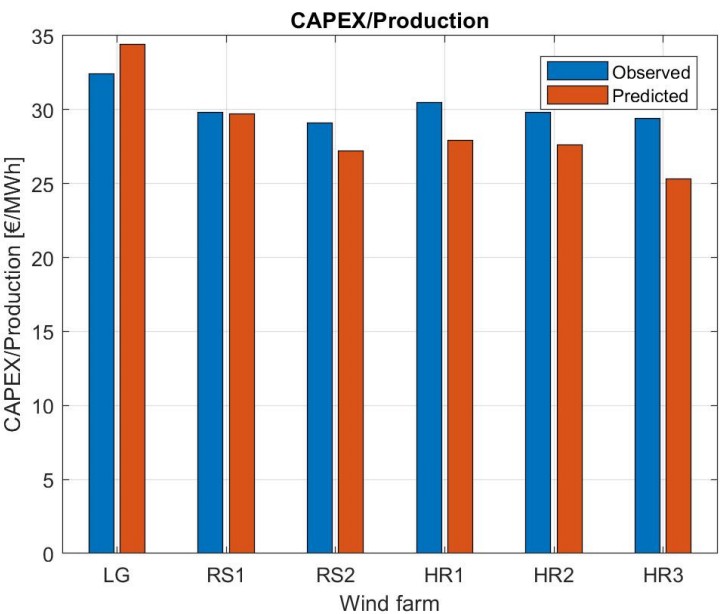

**Figure 7.** Comparison between computed and actual CAPEX to production ratio for the different wind farms.

## 5. Conclusions

A simple framework capturing the essential elements for assessing available wind resources and economic aspects of erecting large offshore wind farms, is developed. This results in a fast and robust minimalistic prediction model, which with a limited number of easily accessible input variables can estimate both the annual energy output and the associated costs for a given offshore wind farm. The wind resources are determined by combining a wake model for large wind farms with a modified Weibull distribution of the average hub height wind speed within the wind farm. The economic analysis includes the main components involved in costs of offshore wind farms, such as the cost of the wind turbine, support structures, OM, and the electric grid.

The model is validated against data from a number of well-documented wind farms, including the Lillgrund wind farm, the Rødsand wind farms, and the Horns Rev wind farms. These data, which cover about two decades of experience, are collected from different sources and are modified in order to refer to 2019 prices. The comparison shows generally a very good agreement between predicted and actual data. For most wind farms, the computed annual production lies within 5% of the actual one, except for the Lillgrund wind farm, which is under-predicted by about 9%. This discrepancy is most likely due to the small size of this wind farm, as the production model assumes equilibrium between the velocity deficit of the atmospheric boundary layer and the momentum impact from the wind farm, which in principle requires an "infinite" wind farm. The difference between the actual and the estimated power production provided by the developer of the Horns Rev 2 wind farm indicates that developers operate with production uncertainties of about 10%, hence the results provided by the simplified model are deemed very good.

The financial cost of erecting the wind farms (CAPEX) is predicted within a maximum error of 10%, with a tendency to under-predict the actual costs. Finally, computed and actual data for CAPEX divided by production over the lifetime of the park (20 years) are in very good agreement for all the wind farms. It is very interesting that the values are nearly identical for the various wind farms, and, as a very good estimate, a unified value of 30 EUR/MWh may be used as a rough and ready number for the CAPEX/production ratio.

**Author Contributions:** The two authors contributed equally to the work and the writing of the manuscript with G.C.L. heading the model formulation and J.N.S. heading the model implementation

as well as the model validation. Both authors have read and agreed to the published version of the manuscript.

**Funding:** This research was funded by of the Danish Council for Strategic Research for the project Center for Computational Wind Turbine Aerodynamics and Atmospheric Turbulence (grant 2104-09-067216/DSF) (COMWIND: http://www.comwind.org) as well as by the EU Horizon 2020 research and innovation program, under grant agreement no. 727680 (TotalControl).

**Institutional Review Board Statement:** Not applicable.

**Informed Consent Statement:** Not applicable.

**Data Availability Statement:** The Maplab script used in the present study is freely available by contacting the authors.

**Conflicts of Interest:** The authors declare no conflict of interest.

## Abbreviations

| | |
|---|---|
| a | Calibration constant |
| $c_t$ | Dimensionless auxiliary parameter |
| f | Coriolis parameter |
| $f(*,*,*)$ | Weibull probability density function |
| $f_C$ | Wind turbine capacity factor |
| $f_S(*)$ | Wind turbine load factor |
| $f_{WF}$ | Wind farm capacity factor |
| $f_{WT}(*|*)$ | Wind turbine size factor |
| h | Hub height |
| k | Weibull shape parameter |
| $u^*_{lo}$ | Friction velocity for the lower part of the boundary layer |
| $u^*_{hi}$ | Friction velocity for the upper part of the boundary layer |
| x | Realization of a stochastic variable X |
| z | Height above sea surface in m |
| $z_{ref}$ | Reference height above sea surface in m |
| $z_{0,lo}$ | Roughness length characteristic for the lower part of the boundary layer |
| $z_{0,hi}$ | Roughness length characteristic for the upper part of the boundary layer |
| A | Wind farm area |
| $A_R$ | Rotor area |
| $C_{add}$ | Additional costs in EUR |
| $C_C$ | Wind farm grid financial costs pr. running meter in EUR |
| $C_G$ | Aggregated internal wind farm grid costs in EUR |
| $C_{FJ}$ | Cost of a jacket support structure in MEUR |
| $C_{FM}$ | Cost of a monopile support structure in MEUR |
| $C_{O\&M}(*,*,*)$ | Cost of operation and maintenance (OM) in EUR |
| $C_{O\&M,base}(*,*)$ | Cost of operation and maintenance (OM) excluding transportation to site in EUR |
| $C_{O\&M,L}(*)$ | Cost of transportation associated with operation and maintenance (OM) in EUR |
| $C_p$ | Planning costs in MEUR |
| $C_P$ | Power coefficient |
| $C_{P,rated}$ | Power coefficient at rated wind speed |
| $C_s$ | Cost of substation in MEUR |
| $C_T$ | Thrust coefficient |
| $C_{total}$ | Total cost of an offshore wind farm installation in MEUR |
| $C_{T,rated}$ | Thrust coefficient at rated wind speed |
| $C_{WT}$ | Cost of a wind turbine in MEUR |
| $C_{WTref}$ | Yearly cost of OM for a reference wind turbine in EUR |
| $C_{yref}$ | Cost of a 20 km export cable in MEUR |
| CAPEX | Total cost of an offshore wind farm installation (i.e., capital expenditures) |
| D | Rotor diameter |
| Dw | Water depth in m |
| E | Annual energy production in MWh |

| $G$ | Geostrophic wind speed |
|---|---|
| Ht | Wind turbine tower height |
| $L_C$ | Aggregated length of internal wind farm grid cables |
| Ls | Average distance from wind farm to the shore |
| $L_T$ | Wind turbine inter spacing |
| $LCOE$ | Levelized cost of energy |
| $N_T$ | Number of wind farm wind turbines |
| $N_Y$ | Life time of the wind farm in years |
| OPEX | Operational expenditures |
| $P(U)$ | Wind turbine power production at mean wind speed $U$ |
| $P_E$ | Average annual wind farm power production |
| $P_g$ | Name plate generator capacity |
| $P_G$ | Generator power |
| $P_{R,ref}$ | Rated power of a reference wind turbine in MW |
| $P_{S,y}$ | Average annual power yield of a solitary turbine in MWh |
| $P_{WF,y}$ | Average annual power yield of a wind farm turbine in MWh |
| $P_y$ | Yearly average production of a wind turbine in MWh |
| PoA | Power density |
| PPA | Power purchase agreement |
| $S$ | Normalized wind turbine inter spacing |
| $U$ | Mean wind speed |
| $U_h$ | Mean wind speed at wind turbine hub height |
| $U_{h,0}$ | Ambient mean wind speed at wind turbine hub height |
| $U_{lo}$ | Mean wind speed at lower part of the boundary layer |
| $U_{hi}$ | Mean wind speed at upper part of the boundary layer |
| $U_{in}$ | Cut-in mean wind speed |
| $U_{out}$ | Cut-out mean wind speed |
| $U_r$ | Rated mean wind speed |
| $X$ | Stochastic variable |
| $Y$ | Distance from site to service harbor in km |
| $Y_{ref}$ | Reference distance from site to service harbor in km |
| $\alpha$ | Auxiliary coefficient |
| $\beta$ | Auxiliary coefficient |
| $\delta$ | Auxiliary parameter |
| $\varepsilon_1$ | Auxiliary parameter |
| $\varepsilon_2$ | Auxiliary parameter |
| $\varphi$ | Latitude |
| $\gamma$ | Auxiliary parameter |
| $\gamma_F$ | Fraction of wind turbines erected on monopole foundations |
| $\kappa$ | von Kármán constant |
| $\lambda$ | Weibull scale parameter |
| $\rho$ | Air density |
| $\tau_w$ | Surface friction stress |
| $\tau_{w,hi}$ | Surface friction stress |
| $\tau_{w,lo}$ | Surface friction stress |
| $\Gamma (*,*)$ | Incomplete Gamma function |
| $\Omega$ | Rotational speed of the earth |

## Appendix A

*Appendix A.1. Average Production under Ambient Flow Conditions*

Ambient wind speed statistics over the year (typically based on 10 or 30 min averaging periods) are traditionally quantified using a two-parameter Weibull distribution. The probability density function (PDF) of a Weibull distributed random variable is

$$f(x; \lambda, k) = \begin{cases} \frac{k}{\lambda} \left( \frac{x}{\lambda} \right)^{k-1} e^{-\left( \frac{x}{\lambda} \right)^k}; & x \geq 0 \\ 0; & x < 0 \end{cases} \tag{A1}$$

where $x$ is a realization of a stochastic variable $X$, $k > 0$ is the Weibull shape parameter, and $\lambda > 0$ is the Weibull scale parameter. The yearly average production of the wind turbine $P_y$ may be formulated as a convolution of the wind turbine production characteristics with the mean wind speed probability density function expressed in Equation (A1). Thus

$$\begin{aligned} P_y &= \int_{U_{in}}^{U_{out}} P(U) f(U; \lambda, k) dU \\ &= \alpha \int_{U_{in}}^{U_r} U^3 f(U; \lambda, k) dU + \beta \int_{U_{in}}^{U_r} f(U; \lambda, k) dU + P_G \int_{U_r}^{U_{out}} f(U; \lambda, k) dU \end{aligned} \tag{A2}$$

where the coefficients $\alpha$ and $\beta$ are determined from the cut-in wind speed and rated conditions, Equations (1) and (2). Reformulating the Weibull distribution, Equation (A1), as

$$f(U; \lambda, k) = \begin{cases} \frac{-d}{dU} e^{-\left( \frac{U}{\lambda} \right)^k}; & x \geq 0 \\ 0; & x < 0 \end{cases}, \tag{A3}$$

Equation (A2) simplifies to

$$P_y = \alpha \int_{U_{in}}^{U_r} U^3 f(U; \lambda, k) dU + \beta \left( e^{-\left( \frac{U_{in}}{\lambda} \right)^k} - e^{-\left( \frac{U_r}{\lambda} \right)^k} \right) + P_G \left( e^{-\left( \frac{U_r}{\lambda} \right)^k} - e^{-\left( \frac{U_{out}}{\lambda} \right)^k} \right). \tag{A4}$$

The remaining integral in Equation (A4) is solved using the variable transformation $t = \left( \frac{U}{\lambda} \right)^k$, whereby we obtain

$$\int_{U_{in}}^{U_r} U^3 f(U; \lambda, k) dU = \lambda^3 \int_{(U_{in}/\lambda)^k}^{(U_r/\lambda)^k} t^{3/k} e^{-t} dt = \lambda^3 \left[ \Gamma \left( \frac{3+k}{k}, \left( \frac{U_{in}}{\lambda} \right)^k \right) - \Gamma \left( \frac{3+k}{k}, \left( \frac{U_r}{\lambda} \right)^k \right) \right], \tag{A5}$$

where $\Gamma(*,*)$ is the incomplete gamma function (cf., Abramowitz and Stegun [10]). Finally, introducing Equation (A5) in Equation (A4), we obtain the following closed form expression for the average wind turbine production:

$$P_y = \alpha \lambda^3 \left[ \Gamma \left( \frac{3+k}{k}, \left( \frac{U_{in}}{\lambda} \right)^k \right) - \Gamma \left( \frac{3+k}{k}, \left( \frac{U_r}{\lambda} \right)^k \right) \right] + \beta \left( e^{-\left( \frac{U_{in}}{\lambda} \right)^k} - e^{-\left( \frac{U_r}{\lambda} \right)^k} \right) + P_G \left( e^{-\left( \frac{U_r}{\lambda} \right)^k} - e^{-\left( \frac{U_{out}}{\lambda} \right)^k} \right) \tag{A6}$$

The Weibull parameters depend in general on altitude as well as on the stability conditions of the ABL. For the present study, we simplify matters by assuming neutral ABL stability conditions "in average", and under this assumption we conjecture that the Weibull shape parameter is independent of altitude. The mean of the Weibull distribution (i.e., the yearly mean wind speed) $U_y$ may be expressed as

$$U_y = \lambda \, \Gamma(1 + 1/k), \tag{A7}$$

where $\Gamma$ (*) is the gamma function. As seen, $U_y$ scales directly with the Weibull scale parameter for a fixed shape parameter. Discharging non-neutral atmospheric boundary layer stability conditions, a logarithmic shear profile may be assumed, meaning that the relative increase in mean wind speed $f_{\Delta U}$ for an increase in altitude from a reference height $z_{ref}$ to height $z$ is given by

$$f_{\Delta U} = U/U_{ref} = Ln(z/z_0)/Ln\left(z_{ref}/z_0\right), \tag{A8}$$

with $z_0$ being the surface roughness length and $U_{ref}$ being the mean wind speed at the reference height.

The wind turbine capacity factor, $f_C$, expresses the ratio of the actual yearly output to its potential output, if it were possible to operate at full nameplate capacity continuously over the year. For a solitary turbine it is accordingly defined as

$$f_C = P_y/P_G, \tag{A9}$$

with $P_y$ obtained from Equation (A6).

Assuming that the Weibull shape parameter is independent of altitude, the formulas for turbine average production (Equation (A6)) and capacity factor (Equation (A9)) apply for all altitudes, if the Weibull scale parameter $\lambda$ associated with the reference height, is replaced with $f_{\Delta U}\lambda$ (cf., Equation (A8)). In the above, the roughness length has implicitly been assumed constant, which strictly speaking is true only for an onshore site. For offshore conditions the surface roughness depends on the wind speed, which complicates matters somewhat. However, this is disregarded in the present study.

*Appendix A.2. Average Production under Wind Farm Flow Conditions*

The wind speed statistics inside a wind farm are different from the wind speed statistics of the ambient undisturbed flow discussed in the previous subsection. This is due to the wind speed reduction caused by the wind turbines, which, for a large wind farm, may be estimated according to Equation (13). Here we derive the distribution of the mean wind speed at hub height inside the wind farm and in turn estimate the average power production of the wind turbines operating inside the wind farm. This estimate will be based on an assumed Weibull distributed ambient mean wind speed at relevant hub heights, meaning that the Weibull scale parameter $\lambda$ may be adjusted by the factor defined in Equation (A8) in case the hub height in question differs from the reference hub height.

We note that the mean wind speeds at hub height inside and outside the wind farm are described by two interrelated stochastic variables. We will consider the mean wind speed inside the wind farm as resulting from a transformation of the ambient undisturbed mean wind speed according to the receipt described in Section 2.1. The mean wind speed at hub height $U_h$ inside the wind farm is given by Equation (13). For $c_t = 0$, we obtain the ambient mean wind speed at hub height as

$$U_{h,0} = \frac{G}{1 + ln\left(\frac{G}{fh}\right)\frac{1}{ln(h/z_0)}}. \tag{A10}$$

We introduce the following short hand notation

$$\gamma = ln\left(\frac{G}{fh}\right), \quad \delta = ln\left(\frac{h}{z_0}\right), \tag{A11}$$

whereby

$$U_h\left[1 + \gamma\frac{\sqrt{c_t + (\kappa/\delta)^2}}{\kappa}\right] = U_{h,0}\left[1 + \frac{\gamma}{\delta}\right], \tag{A12}$$

or

$$U_h = U_{h,0} \frac{1 + \frac{\gamma}{\delta}}{1 + \gamma \frac{\sqrt{c_t + (\kappa/\delta)^2}}{\kappa}}. \tag{A13}$$

Introducing Equation (12) into Equation (A13), one obtains

$$\frac{U_h}{U_{h,0}} = \begin{cases} \varepsilon_1 = \dfrac{1 + \frac{\gamma}{\delta}}{1 + \frac{\gamma}{\kappa}\sqrt{\frac{\pi C_{T,rated}}{8S^2} + (\kappa/\delta)^2}}; \ U_{in} \leq U_h < U_r \\[4mm] \varepsilon_2 = \dfrac{1 + \frac{\gamma}{\delta}}{1 + \frac{\gamma}{\kappa}\sqrt{\frac{\pi C_{T,rated}}{8S^2}(U_r/U_h)^{3/2} + (\kappa/\delta)^2}}; \ U_r \leq U_h \leq U_{out} \end{cases} \tag{A14}$$

As seen from Equation (A14), $\varepsilon_1$ is a constant whereas $\varepsilon_2 = \varepsilon_2(\overline{U}_h)$ depends on the actual wind speed at hub height.

To determine the probability density function for the wind climate inside the wind farm, we exploit the following relationship between the original Weibull distribution $f_{h,0}$ and the altered distribution $f_h$ due to the wake effects from the wind turbines in the farm:

$$f_h(U_h)dU_h = f_{h,0}(U_{h,0})dU_{h,0}. \tag{A15}$$

The probability density function of $U_H$ in the below rated regime can now be formulated in closed form by combining Equations (A14) and (A15):

$$f_h(U_h) = f_{h,0}(U_{h,0})\frac{dU_{h,0}}{dU_h} = \frac{f_{h,0}(U_h/\varepsilon_1)}{\varepsilon_1}; \ \ U_{in} \leq U_h < U_r \tag{A16}$$

It is intuitively clear that, with the wind speed transformation expressed in Equation (A14) for the below rated regime, an infinitesimal probability around $U_{h,0}$ for the ambient conditions, equals an infinitesimal probability around $U_h$ for the infinite wind farm conditions, which is exactly what is expressed in Equation (A16). As in Appendix A.1, we assume the ambient mean wind speeds to be Weibull distributed (cf., Equation (A1)), whereby we finally obtain the following mean wind speed probability density function for the below rated wind farm wind climate:

$$f_h(U_h) = f_{h,0}(U_h; \varepsilon_1\lambda, k); \ U_{in} \leq U_h < U_r, \tag{A17}$$

which is a Weibull distribution with scale parameter $\varepsilon_1\lambda > 0$.

We now turn to the above rated wind farm regime. Assuming again that the mean wind speed in the ambient flow domain is Weibull distributed, the expected yearly wind turbine production for the above rated wind farm wind speed regime may be formulated as

$$P_G \int_{U_r}^{U_{out}} f_h(U_h)dU_h = P_G \int_{U_r/\varepsilon_2(U_r)}^{U_{out}/\varepsilon_2(U_{out})} f_{h,0}(U_{h,0}; \lambda, k)dU_{h,0}. \tag{A18}$$

or, using Equation (16)

$$P_G \int_{U_r}^{U_{H,o}} f_h(U_h)dU_h = P_G \left( e^{-\left(\frac{U_r}{\varepsilon_2(U_r)\lambda}\right)^k} - e^{-\left(\frac{U_{out}}{\varepsilon_2(U_{out})\lambda}\right)^k} \right). \tag{A19}$$

Employing Equation (A19), and otherwise taking a similar approach as the one leading to Equation (A6) for a solitary wind turbine, the yearly power output is determined as

$$P_{WF,y} = \int\limits_{U_{in}}^{U_{out}} P(U_h) f_h(U_h; \varepsilon\lambda, k) dU_h$$

$$= \alpha \int\limits_{U_{in}}^{U_r} U_h^3 f_h(U_h; \varepsilon_1\lambda, k) dU_h + \beta \int\limits_{U_{in}}^{U_r} f_h(U_h; \varepsilon_1\lambda, k) dU_h + P_G \left( e^{-\left(\frac{U_r}{\varepsilon_2(U_r)\lambda}\right)^k} - e^{-\left(\frac{U_{out}}{\varepsilon_2(U_{out})\lambda}\right)^k} \right) \tag{A20}$$

The first two terms in Equation (A20) can be determined analytically, in analogy with the derivation leading to Equation (A6), and we thus finally obtain the following closed form expression for the average annual power output of a wind farm turbine:

$$
\begin{aligned}
P_{WF,y} = {}& \alpha(\varepsilon_1\lambda)^3 \left[ \Gamma\left(\frac{3+k}{k}, \left(\frac{U_{in}}{\varepsilon_1\lambda}\right)^k\right) - \Gamma\left(\frac{3+k}{k}, \left(\frac{U_r}{\varepsilon_1\lambda}\right)^k\right) \right] + \beta \left( e^{-\left(\frac{U_{in}}{\varepsilon_1\lambda}\right)^k} - e^{-\left(\frac{U_r}{\varepsilon_1\lambda}\right)^k} \right) \\
& + \quad P_G \left( e^{-\left(\frac{U_r}{\varepsilon_2(U_r)\lambda}\right)^k} - e^{-\left(\frac{U_{out}}{\varepsilon_2(U_{out})\lambda}\right)^k} \right)
\end{aligned}
\tag{A21}
$$

Equations (A15) and (A16) result from considering a transformation, given by Equation (A22), between the two stochastic variables $U_h$ and $U_{h,0}$. A precondition for obtaining the simple degenerated expressions resulting from this transformation, given by Equations (A15) and (A16), is that $U_h = U_h(U_{h,0})$ is a monotonic function. For the below rated wind speed case this is easily shown as $\varepsilon_1$ in Equation (A16) is a constant. For the above rated wind speed case a formal proof is given in Appendix B.

### Appendix B

In this appendix, the gradient of the wind farm mean wind speed $U_h$ with respect to the ambient mean wind speed $U_{h,0}$ is proven to be positive in the above rated wind speed regime. From Equation (A14) we have

$$U_h = U_{h,0} \frac{1 + \frac{\gamma}{\delta}}{1 + \frac{\gamma}{\kappa}\sqrt{\frac{\pi C_{T,rated}}{8S^2}(U_r/U_h)^{3/2} + (\kappa/\delta)^2}}, \tag{A22}$$

or

$$U_{h,0} = U_h \left(1 + \frac{\gamma}{\delta}\right)^{-1} \left(1 + \frac{\gamma}{\kappa}\sqrt{\frac{\pi C_{T,rated}}{8S^2}(U_r/U_h)^{3/2} + (\kappa/\delta)^2}\right). \tag{A23}$$

The gradient is thus expressed as

$$
\begin{aligned}
\frac{dU_{h,0}}{dU_h} = {}& \left(1 + \frac{\gamma}{\delta}\right)^{-1} \left(1 + \frac{\gamma}{\kappa}\sqrt{\frac{\pi C_{T,rated}}{8S^2}(U_r/U_h)^{3/2} + (\kappa/\delta)^2}\right) \\
& + U_h \left(1 + \frac{\gamma}{\delta}\right)^{-1} \times \left(\frac{3}{4}\frac{\gamma}{\kappa}\frac{\pi C_{T,rated}}{8S^2}(U_r/U_h)^{1/2}\left(\frac{U_r}{U_h^2}\right) \times \left(\frac{\pi C_{T,rated}}{8S^2}(U_r/U_h)^{3/2} + (\kappa/\delta)^2\right)^{-1/2}\right)
\end{aligned}
\tag{A24}
$$

With $\gamma$, $\kappa$, $\delta$, $U_r$, and $U_H$ being positive, $dU_{h,0}/dU_h$ is positive, and thereby $dU_h/dU_{h,0}$ is positive for any (positive) value of $U_{h,0}$, which in turn means that $U_h(U_{h,0})$ is strictly monotonic. As seen, this qualitative result has been obtained without knowing the explicit form of the function $U_h(U_{h,0})$.

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
