# Peer review of "A Minimalistic Prediction Model to Determine Energy Production and Costs of Offshore Wind Farms"

_energies, doi:10.3390/en14020448_

Round 1

Reviewer 1 Report

This study deals with interesting issue of modeling the production and costs of wind farms. Authors proposed simple algorithm to predict the productivity of offshore wind farms and confirm that, this model have good accuracy.

The paper was well organized. However I found some issues that nedd to be claryfied/improved:

  • The more recent publication should be reviewed in the introduction, 
  • please define all symbols used in equations,
  • line 112, there is some problem with equation for Cp, what is "?" used in the equation?
  • There is problem with equations 5, 6, 14, 17, 28, 29, A1, A2, A3, A17,  there are strange symbols, without this correction I can not veryfie the corecctnes of the proposed methods,
  • Please add the reference sources for equations where appropriate, 
  • Please show the sorce for the data used in the comparison of prediction results and actual data from wind turbine, from where came from the data for wind turbines, are they available for readers? if yes please give appropriate link, citation.
  • Please check and improve language in entire manuscript.

Author Response

Please see the response in the attachment.

Reviewer 2 Report

The manuscript deals with a topic within the scopes of Energies journal. It is written in sufficient English language.

I have some major concerns regarding the significance and the novelty of the manuscript. To be honest, the paper must be completely rewritten in order to follow a track that starts from the problem that is tackled by the authors to the solution proposed by the authors.

1) Much material of Section 2 is redundant and comes from well known equations about wind energy. This material could be removed without any detrimental effect.
2) The authors must clearly state the novel contribution brought by their research, which is lost among known material. The introduction also fails to individuate any novelty brought by this manuscript. To be honest, I still have only a vague idea about what does this paper aim to solve. 
3) The cost equations are based on too simplifying assumptions. For example, the equation used to model the cost of a wind turbine in Section 2.6.1. appears to be very simple and it cannot cover any rated power; also, it does not account for scaling power, as it is linear in Prated. 
How does this simple modeling of the costs impact on the outcomes?
4) The results are not clearly presented, and this keeps failing to individuate the scope of the manuscript. The results deal with a "prediction" of the generation, but this is not prediction, rather it is characterization of the wind generation as no forecasts are involved. The cost analysis fails to evidence any relevant information for the reader.

Minor comments:

- There are many strange symbols occurring in the equations (e.g., 5, 6, 14, 28, 29...)
- line 112: there is a "?" symbol in the CP equation

Reviewer 3 Report

GENERAL COMMENTS

The work is interesting, it has a clear degree of originality, and it might be appropriate for publication in Energies journal, after performing a major and very careful revision.

Indeed the topic targeted is very important since offshore wind energy is one of the most important renewable energy resources and thus represents an issue of increasing importance. However, the proposed work should be still completed and improved in various points in order to be ready for a journal publication. Furthermore, now the proposed work looks more like a technical report rather than a scientific paper.

Some of the changes required are:

- More information and arguments concerning the validity and limitations of the proposed method should be provided;

- Many equations should be reviewed and rewritten;

- Some sentences should be rephrased in order to express in a clearer way the ideas;

- The general clarity of the work should be improved;

- An additional check of the English grammar and spelling can be done;

- More discussion of the results should be provided.

Some specific comments are given next. They are not exhaustive, which means that there might be also still some other issues to be double-checked by the authors before resubmission.

SPECIFIC COMMENTS

ENGLISH LANGUAGE

This is in general OK. However, an additional English check should be performed since there are still some sentences that need to be reformulated in order to present in a clearer way the ideas and the findings of the proposed work.

KEYWORDS

You repeat three times offshore wind. You should better include offshore wind as a separate keyword and remove it from the other three.

SYMBOLS AND EQUATIONS

There are 37 equations provided and not all seem to be OK. Please check carefully whether all the quantities involved are properly defined. For example, even in equation 1, the meaning of the coefficients alfa and beta is not explained. Furthermore, in the definition of the power coefficient, you have a question mark which is probably a mistake. Also, this expression can be defined as a separate equation.

On the other hand, some of the equations from the main text and all equations provided in Appendices should be reviewed because strange symbols occur in many of them. For example, equations 5, 6, 14, 15, 16, 17, 28, 29, A1, A3, A5, A6, A7, A17, and A21 should be rewritten.

ABBREVIATIONS

Please check carefully if all the abbreviations and notations considered in the work are explained for the first time when they are used, even if these are considered trivial by the authors. The paper should be accessible to a wide audience. For example, the meanings of DKK, O&M, etc. are not explained. Furthermore, an index of notations (and eventually of abbreviations) will be beneficial for a better understanding of the present work.

FIGURES & TABLES

Some corrections are required in relationship with the figures and the figures captions, as follows:

Figure 2 – you should indicate wherefrom is the map, it would make sense to write the longitude and latitude on the figure axes; Also some explanations can be moved from the figure caption to the main text.

The work presents a lot of information. However, more discussions on the data presented should be provided.

MINOR ISSUES

Line 11 (Abstract) – you should better replace ‘wind power’ with ‘wind farm’.

Line 22 (Abstract) – you should better replace ‘date’ with ‘data’.

Please check similar aspects all along with the work.

Round 2

Reviewer 1 Report

Dear Authors,

thank you very much for provided changes. They improve the manuscript quality. However still I have one remark - still the strange symbols in equations are present. Please revise it because it is hard to follow the proposed formulas.

Author Response

We really regret this problem. Our own Words and pdf files are fine, but we can see that after submission the equations in the pdf-file are changed and polluted with strange symbols. However, the uploaded Words file is fine and the equations can be found there.

Reviewer 2 Report

I have no further comments

Author Response

Thats for the comprehensive review that was a great help in ameliorating the quality of the paper.

Reviewer 3 Report

Most of my suggestions have been considered and corrections have been operated. However, there are still some minor issues that should be still corrected:

  • The authors say that ‘a list of symbols and abbreviations is now included’, but I was not able to see where this list is. Please check and insert it. Furthermore, there are still some abbreviations used for which I have not seen the explanation (as for example DTU, WAsP, IEA, ECN, TNO, EWEA);
  • There are still equations for which strange signs occur in the pdf version (see for example eqs: 3, 5, 6, 14, 15, 16, 17, 28, 29, A1, A3, A5, A7, A17, A21). Please check in the PDF version submitted and correct

Author Response

The list of symbols was attached as supplementary material. We have now also included it direcly in theend of the manuscript. The abbreviations are now explained directly on the paper the first time they are used. Most of them are only used one time and we therefore have not included them in the list of symbols.

We really regret this problems with change of symbols. Our own Words and pdf files are fine, but we can see that after submission the equations in the pdf-file are changed and polluted with strange symbols. However, the uploaded Words file is fine and the equations can be found there.